



The importance of non-stationary multiannual periodicities in the NAO index for forecasting
water resource extremes
William Rust [a]; John P Bloomfield [b]; Mark Cuthbert [cd]; Ron Corstanje [e]; Ian Holman [a]
a Cranfield Water Science Institute (CWSI), Cranfield University, Bedford MK43 0AL
b British Geological Survey, Wallingford, OX10 8BB
c School of Earth and Environmental Sciences, Cardiff University, Park Place, Cardiff, CF10
3AT
d School of Civil and Environmental Engineering, The University of New South Wales,
Sydney, Australia
e Centre for Environment and Agricultural Informatics, Cranfield University, Bedford MK43
0AL
Correspondence to Ian Holman (i.holman@cranfield.ac.uk)
**Abstract**
Drought forecasting and early warning systems for water resource extremes are increasingly
important tools in water resource management, particularly in Europe where increased
population density and climate change are expected to place greater pressures on water
supply. In this context, the North Atlantic Oscillation (NAO) is often used to indicate future
water resource behaviours (including droughts) over Europe, given its dominant control on
winter rainfall totals in the North Atlantic region. Recent hydroclimate research has focused
on the role of multiannual periodicities in the NAO in driving low frequency behaviours in
some water resources, suggesting that notable improvements to lead-times in forecasting
may be possible by incorporating these multiannual relationships. However, the importance
of multiannual NAO periodicities for driving water resource behaviour, and the feasibility of
this relationship for indicating future droughts, has yet to be assessed in the context of
known non-stationarities that are internal to the NAO and its influence on European
meteorological processes. Here we quantify the time-frequency relationship between the
NAO and a large dataset of water resources records to identify key non-stationarities that
have dominated multiannual behaviour of water resource extremes over recent decades.
The most dominant of these is a 7.5-year periodicity in water resource extremes since
approximately 1970 but which has been diminishing since 2005. Furthermore, we show that



the non-stationary relationship between the NAO and European rainfall is clearly expressed
at multiannual periodicities in the water resource records assessed. These multiannual
behaviours are found to have modulated historical water resource anomalies to an extent
that is comparable to the projected effects of a worst-case climate change scenario.
Furthermore, there is limited systematic understanding in existing atmospheric research for
non-stationaries in these periodic behaviours which poses considerable implications to
existing water resource forecasting and projection systems, as well as the use of these
periodic behaviours as an indicator of future water resource drought.

## 1. Introduction

Oscillatory ocean-atmosphere systems (such as El Nino Southern Oscillation (ENSO), North
Atlantic Oscillation (NAO) and Pacific Decadal Oscillation (PDO)) are known to modulate
hydrometeorological processes over a large domain, often driving multiannual periodicities in
hydrological records (Kuss and Gurdak, 2014; Labat, 2010; Trigo et al., 2002). As such,
indices of these systems can be useful when explaining decadal-scale variations in water
resource behaviour in Europe (Svensson et al, 2015; Kingston et al, 2006), North America
(Coleman and Budikova, 2013) and Asia (Gao et al, 2021). In the North Atlantic region, the
NAO represents the principal mode of atmospheric variability and is a leading control on
European winter rainfall totals (Hurrel, 1995; Hurrel and Deser, 2010). As such, many
studies have found strong and significant relationships between the winter NAO Index
(NAOI) and hydrological variables across Europe (Wrzesinski and Paluszkiewicz, 2011;
Brady et al, 2019; Burt and Howden, 2013), leading to the development of seasonal and
long-lead forecasting systems of hydrological behaviour (Svensson et al, 2015, Bonaccorso
et al, 2015).
A growing number of studies have identified stronger relationships between the NAOI and
certain water resource variables at multiannual periodicities (Holman et al, 2011; Neves et



al, 2019; Uvo et al, 2021), than at an annual scale. This is particularly apparent where longer
hydrological response times predominate (Rust et al 2021a). For instance, Neves et al
(2019) identified significant relationships between the NAOI and groundwater level in
Portuguese aquifers and at approximately 6- and 10-year periodicities, with associations to
episodes of recorded groundwater drought. Furthermore, Liesch and Wunsch (2019) found
significant coherence between NAOI and groundwater level at approximately 6- to 16-year
periodicities across the UK, Germany, Netherlands and Denmark. Rust et al (2019; 2021a)
identified a similar significant 6- to 9-year cycle across a large dataset of groundwater level
(59 boreholes) and streamflow (705 gauges) in the UK, which was associated with the
principal periodicity of the NAO (of a similar length (Hurrell et al., 2003; Zhang et al., 2011)).
In the instance of groundwater level, this periodicity was found to represent a notable portion
of overall behaviour (40% the standard deviation), and minima in the cycle were shown to
align with recorded instances of wide-spread groundwater drought (Rust et al, 2019). Given
their association with recorded droughts across Europe, these studies highlight the potential
benefit of an *a priori* knowledge of multiannual NAO periodicities in water resources for
improving preparedness for water resource extremes in Europe. Here we use extremes to
describe water resource deficit (i.e., drought) and periods of anomalously high water
resource stores. This is distinct from hydrological extremes, which infers the drought – flood
continuum.
However, the value of a multiannual relationship between the NAO and European water
resources has yet to be assessed in the context of reported non-stationarities in
hydroclimate systems. For instance, the NAO is an intrinsic mode of atmospheric variability
(Deser et al, 2017), but can also be influenced by multiple other teleconnection systems
such as the Madden-Julien Oscillation, Quasi-Biennial Oscillation (Feng et al 2021) or El-
Nino Southern Oscillation (Zhang et al, 2019). As such it is currently unclear whether
periodicities in the NAOI are emergent behaviours or the result of external forcing. This has
been compounded by a relatively weak signal-to-noise ratio for NAO periodicities, making



confident multiannual signal detection difficult (O'Reilly et al, 2018; Hurrel et al, 1997). While
stronger NAO-like multiannual periodicities have been detected in water resource variables,
due to the high-band filtering function of hydrological processes (van Loon, 2013), the
degree to which these behaviours are sufficiently stable to enable development of predictive
utilities is currently unclear. Furthermore, existing research has shown that the sign of the
relationship between NAOI and European rainfall is non-stationary at decadal timescales
(Rust et al, 2021b); Vicente-Serrano and López-Moreno (2008)). This is expected to add a
degree of uncertainty to the detection of lead times between multiannual periodic
components in the NAO and water resource response, which is necessary in the
development of early warning systems for water resource extremes. While some studies
have ascribed lags to this multiannual relationship for European water resources (Neves et
al, 2019; Holman et al, 2011), the extent to which this non-stationarity is present at
multiannual periodicities has yet to be assessed.
Finally, a critical application of early warning systems for water resource extremes is in the
design of drought management regimes for existing and projected climate change (Sutanto
et al, 2020). While some studies have quantified the degree of modulation that multiannual
ocean-atmosphere systems can have on water resources (Kuss and Gurdak, 2014; Neves et
al., 2019; Velasco et al., 2015), few have compared these to the expected modulations from
projected climate change scenarios. As such the benefit of incorporating multiannual NAO
periodicities into early warning systems for improving preparedness for water resource
extremes in climate change scenarios has not been assessed.
The aim of this paper is to assess the utility of multiannual relationships between the NAO
and water resources for improving preparedness for future water resource extremes. This
aim will be met by addressing the following research objectives:

1.  Quantify significant covariances between multiannual periodicities in the NAOI and

water resource extremes, and assess the extent to which these periodicities are

stable over time





2. Assess multiannual periodicity phase differences between the NAOI and water

resources over time, to understand the extent to which annual-scale non-

stationarities between the NAO and European rainfall are expressed at multiannual

scales

3. Quantify the modulations of water resource variables caused by key multiannual

periodicities in the NAO, during the dry season, and compare this with projected

modulations of water resources due to climate change.

These objectives will be implemented on UK water resource records, given the considerable
coverage of recorded water resource data in time and across the space (Marsh and
Hannaford, 2008); however, the methodologies developed can be applied to any regions.

### 123 2. Data

2.1.   Water resource data
The National Groundwater Level Archive (NGLA) and National River Flow Archive (NRFA)
provide high-resolution spatiotemporal coverage of groundwater level records and
streamflow across the UK.
2.1.1. Groundwater data
Monthly NGLA groundwater level data from 136 boreholes covering all of the major UK
aquifers, with record lengths of more than 20 years and data gaps no longer than 24 months,
have been used (Figure 1). While some meta-analysis was conducted on monthly data, the
primary analysis was undertaken on seasonally averaged data, meaning a data gap of no
more than two points. They cover a range of unconfined and confined consolidated aquifer
types and have been categorised into generalised aquifer groups of Chalk (78 sites),
Limestone (12 sites), Oolite (12 sites), Sandstone (34) and variably cemented mixed clays
and sands (Lower Greensand Group, Allen et al., 1997) (3 sites). Given the spatially
heterogenous response of the Chalk aquifer to droughts (Marchant and Bloomfield, 2018),





Chalk sites have been subdivided into four groups based on aquifer region: Lincolnshire basin
(8 sites), East Anglian basin (17 sites), Thames and Chiltern basin (29 sites) and Southern
basin (21 sites) (Allen et al., 1997; Marchant and Bloomfield, 2018).
Broad aquifer groups can be described as follows: Chalk, a limestone aquifer comprising of a
dual porosity system with localized areas where it exhibits confined characteristics;
characterised by fast-responding fracture porosity (Bloomfield, 1996); Oolite characterised by
a highly fractured lithology with low intergranular permeability; Sandstone, comprised of sands
silts and muds with principle inter-granular flow but fracture flow where fractures persist; and
Lower Greensand, characterised by intergranular flow with lateral fracture flow depending on
depth and formation (Allen et al, 1997).
2.1.2.  Streamflow data
Monthly streamflow data from the UK National River Flow Archive (NRFA; Dixon et al., 2013:
http://nrfa.ceh.ac.uk/) has been used. Gauging stations with more than 20 years of continuous
streamflow data and no data gaps greater than 24 months were initially selected. Sites serving
the largest catchment were selected where there are multiple sites within a single river
catchment. This produced a final list of 767 streamflow gauging stations for use. To
understand broad spatial relationships across the streamflow dataset, records have been
divided into groups based on the NRFA river drainage basin (RDB). These are grouped by
seven generalised regions of the UK; North and West Scotland (75 records), East Scotland
(89 records), Northern Ireland (38 records), North-west England (70 records), North-east
England (102 records), Wales & South-west England (170 records), East Anglia & South-east
England (223 records). Streamflow with minimal influence from human factors is often used
in hydroclimate studies to avoid confounding mechanisms, however no such large-scale
dataset exists for the UK. Furthermore, over the period of analysis and the broad scale of this
assessment, inconsistences in the way water resource management practices are
implemented is expected to result in noise to the observations rather than some systematic
signal or bias that would affect the results of this paper.



## 2.2.    North Atlantic Oscillation data
Monthly North Atlantic Oscillation Index (NAOI) data calculated by the National Centre for
Atmospheric Research (NCAR) using the principal component (PC) method for the period
1989 – 2021 has been used. The PC NAOI is a time series of the leading empirical orthogonal
functions (EOFS) of sea level pressure grids across the north Atlantic region (20°-80°N, 90°W-
40°E).

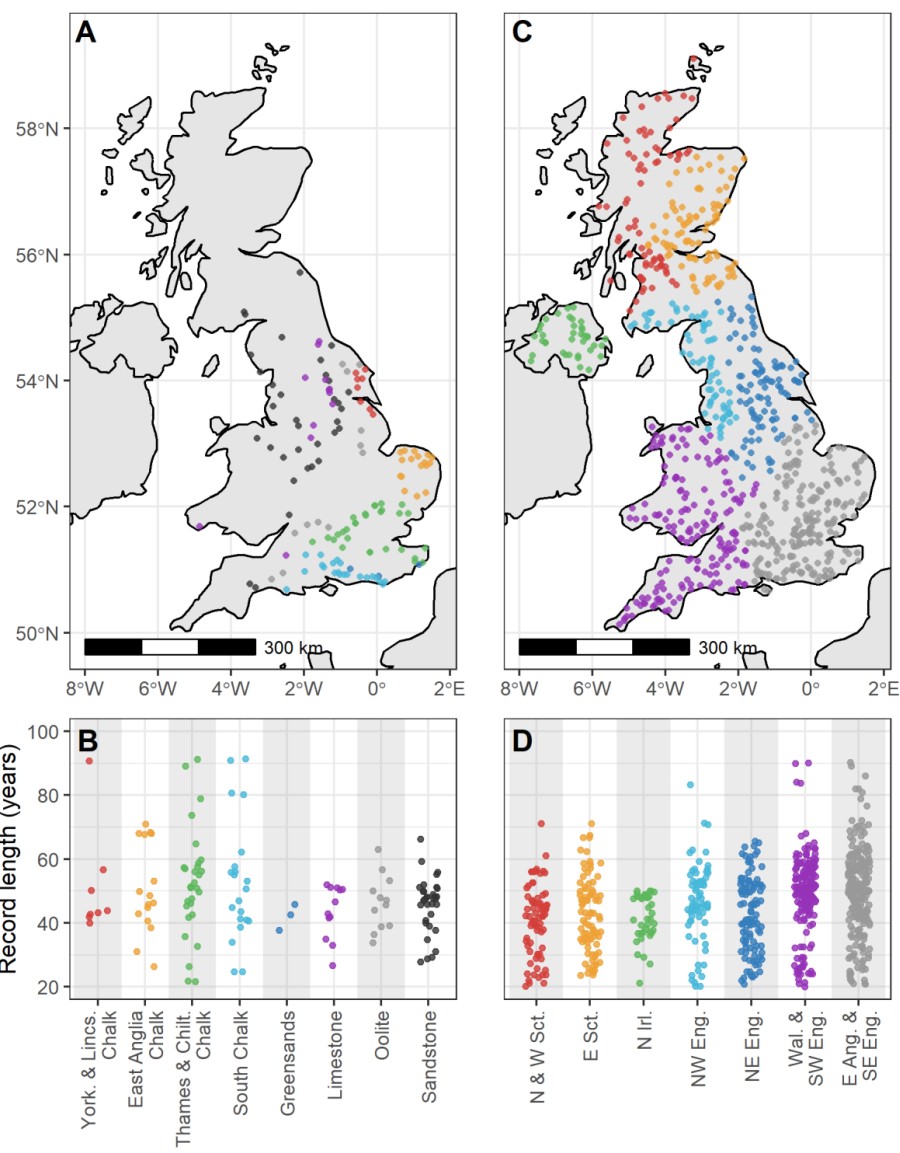


Figure 1 – Spatial and temporal distributions of water resource records; a) location of groundwater boreholes coloured by associated aquifer group, b) jitter plot of groundwater record lengths within each aquifer group, c) location of streamflow gauges coloured by associated regional group, d) jitter plot of streamflow record lengths within each regional group

176

177



## 3. Methods

### 3.1. Data Pre-processing

In this study we use the continuous and cross-wavelet transform to understand behaviours and relationships across different periodicities within the different water resource variable time series.

For all datasets, gaps less than two years were infilled to a monthly time step using a cubic spline to produce a complete time series for the wavelet transform. For time series with gaps greater than two years, the shortest time period before or after the data gap was removed. The records were not trimmed to obtain a common period of data coverage. Instead, all data was trimmed to start at a minimum of 1930. This was to allow the analysis of the fewer records that cover a longer time period while still capturing a time periods with adequate record coverage. All of the time series were standardised by dividing by their standard deviation and subtracting their mean.

### 3.2. Quantifying wide-spread water resource extremes

In order to meet objective 1, we produced a time series which describes the behaviour of wide-spread water resource extremes across each resource variable (i.e., groundwater or streamflow). In this study we have assessed water resource extremes using a drought threshold methodology proposed in Peters (2003). While other measures of drought are available (e.g., Standardised Precipitation Index (SPI) and Standardised Groundwater Index (SGI)) (Bloomfield and Marchant, 2013), a threshold approach has been adopted as its can be easily applied to both streamflow and groundwater variables.

To calculate a drought series from monthly groundwater level and streamflow series, we first used the threshold methodology given by equation 4.3 in Peters (2003):

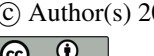



$$\int_0^M (x_t(c) - x(t))_+ \, dt = c \int_0^M (\bar{x} - x(t))_+ \, dt \qquad \text{(Eq. 5)}$$

Where:
$$x_+ = \begin{cases} x & if \ x \geq 0 \\ 0 & if \ x < 0 \end{cases}$$

and M is the full length of the data series. Here we use a threshold level of c = 0.3 for
groundwater level and c = 0.01 for streamflow. Peters et al (2003) found that a value of 0.3
for groundwater level was comparable to other commonly used thresholds. A value of 0.01
for streamflow was chosen as it produced a similar distribution of drought events as the
groundwater drought series. The chosen value of c for either variable is not expected to
affect the outcomes of the study as the focus is on the frequency structure of water resource
extremes, rather than magnitude.
For each measurement site, the monthly time series of drought status (whether in drought
according to the threshold criteria or not) was converted into a yearly series describing
whether that site experienced a drought in the calendar year. Then, for each year, the
number of sites that experienced drought were summed and divided by the number of sites
with coverage of that year. This produced a time series of the proportion of sites
experiencing drought each year, for groundwater level and streamflow variables. This is
referred to as the drought coverage time series.
**3.3. Frequency Transformations**
**3.3.1. Continuous Wavelet Transform (CWT)**
The Continuous Wavelet Transform (CWT) was performed on the drought coverage time
series for groundwater and streamflow to understand the frequency behaviour of wide-
spread water resource extremes over time. The CWT is often used in geoscience to
understand non-stationarities of a variable over time and frequency space (Sang, 2013).





The cross-wavelet transform, $W$, consists of the convolution of the data sequence ($x_t$) with
scaled and shifted versions of a mother wavelet (daughter wavelets):

$$W(\tau, s) = \sum_t x_t \frac{1}{\sqrt{s}} \psi * \left( \frac{t - \tau}{s} \right) \qquad \text{(Eq. 1)}$$

where the asterisk represents the complex conjugate, $\tau$ is the localized time index, $s$ is the
daughter wavelet scale and $dt$ is increment of time shifting of the daughter wavelet. The
choice of the set of scales $s$ determines the wavelet coverage of the series in its frequency
domain. The Morlet wavelet was favoured over other candidates due to its good definition in
the frequency domain and its similarity with the signal pattern of the environmental time
series used (Tremblay et al. 2011; Holman et al. 2011).
The modulus of the transform can be interpreted as the continuous wavelet power (CWP):

$$P(\tau, s) = |W(\tau, s)| \qquad \text{(Eq. 2)}$$

We use the package "WaveletComp" produced by Rosch & Schmidbauer (2018) for all
wavelet transformations in this paper.
**3.3.2. Cross-Wavelet Transform (XWT)**
The bivariate XWT was applied between the NAOI and each of the water resources records
(groundwater level (GWL) and streamflow (SF)). This produces a cross-wavelet power which
is analogous to the covariance between the two variables over a time and frequency
spectrum. This has been selected over the cross-wavelet coherence (analogous to
correlation) as this metric requires a high degree of spectral smoothing, making the resultant
coherence spectra sensitive to the choice of smoothing approach (Rosch & Schmidbauer
(2018))Here we use the covariance spectrum to compare against the drought series
frequency spectrum to understand where strong coherences are reflective of dominant
behaviours in water resource extremes.





In order to calculate cross-wavelet power (XWP) for the bivariate case, it is first necessary to
calculate the continuous wavelet transform (CWT) for each of the variables separately. The
XWT between variables x and y is given by:

$$W.xy(\tau, s) = \frac{1}{s} \cdot W.x(\tau, s) \cdot W.y*(\tau, s)$$     (Eq. 3)

The modulus of the transform can be interpreted as the cross-wavelet power (XWP):

$$P.xy(\tau, s) = |W.xy(\tau, s)|$$     (Eq. 4)


### 3.3.3.  Wavelet Significance

Lag-1 autocorrelations (AR1) in environmental datasets can produce emergent low frequency
behaviours, making the detection of externally-forced behaviours more difficult (Allen and
Smith, 1996; Meinke et al., 2005; Velasco et al., 2015). In this study, a significance test was
undertaken to test the red-noise null hypothesis that wavelet powers calculated are the result
of the recorded variables' AR1 properties. This was based on 1000 synthetic Monte Carlo
series with the original AR1 values. In this paper we test significance to the 95% CI.
The significance spectra for the XWT for each variable pair (e.g., GWL and NAOI) form the
primary results for the XWT method in this paper, since the cross-wavelet power is heavily
dependent on the individual series and its frequency composition. The overall relationship
between the NAOI and water resources as a whole are investigated by showing the proportion
of sites over time and frequency that exhibit a significant relationship with the NAOI (95% CI).
This average significance spectrum is produced by summing the significance matrices across
each resource (groundwater level or streamflow) and dividing by the number of records used
in year each.

### 3.3.4.  Phase Difference





In the bivariate case, the instantaneous phase difference for the XWP spectrum (between
wavelets pairs from the CWT spectrum for each variable) can also be calculated as:

$$Angle(\tau, s) = Arg(W.xy(\tau, s)) \qquad \text{(Eq. 5)}$$


This is the difference of the individual phases from both variables at an instantaneous time
and frequency (period), converted to an angle between $-\pi$, and $\pi$. Values close to 0 indicate
the two series move in-phase, with absolute values close to $\pi$ indicating an out-of-phase
relationship. Values between 0 and $\pi$ indicate degrees of phase difference or phase shift.
Phase differences between 0 and $\pi$ can indicate the degree to which variable x is leading
variable y, however a phase difference between 0 and $-\pi$ can either indicate that variable y is
leading variable x, or that variable x is leading by more than half the phase rotation (period
length). The degree to which a certain variable is leading is analogous to a lag between the
two variables.

**3.4.    Modulation measurement**
In order to understand the degree of modulation that the NAO teleconnection has on water
resources, an absolute and relative modulation value has been calculated for each series.
Here, we use modulation to describe the degree to which the NAO (or other process) has
increased or decreased a water resource measure from its mean. This has been derived by
reconstructing a specific principal periodicity range from the cross-wavelet powers using the
following equation:

$$(x_t) = \frac{dj \cdot dt^{1/2}}{0.776 \cdot \psi(0)} \sum_s \frac{Re(W(.,s))}{s^{1/2}} \qquad \text{(Eq. 6)}$$

Where dj is the frequency step and dt is the time step.



This produces a periodic reconstruction of a component of the original dataset that conforms
to the set of periodicities (scale steps) selected. The mean and maximum amplitude of this
periodic reconstruction was calculated from the absolute values of minima and maxima.
Since the data were standardised by dividing by the standard deviation prior to the wavelet
transform, this calculated mean and maximum amplitude are also relative to the sd of the
original data. Multiplying the calculated amplitude by the original sd converts this back into a
real-valued measurement. This was only done for groundwater, since streamflow is highly
dependent on catchment size. In the case of streamflow, amplitudes are reported as relative
to the standard deviation of the streamflow record. All calculated modulations were produced
using reconstructed wavelets from after 1970 where the majority of records are present in
both groundwater and streamflow variables. This was done to mitigate the effect of differing
record lengths.

**4. Results**
**4.1.    Multiannual water resource extremes covariance with NAOI**
Figure 2 shows the NAOI covariance significance spectrum (fig 2a and 2b) and drought
frequency spectrum (fig 2c and 2d) for the groundwater level records. These have been
plotted together to allow for easier interpretation and comparison of the results. Black lines in
the spectral plots show the 95% CI. The calculated drought series (fig 2e) and record
coverage (fig 2f) have also been plotted alongside for comparison.
Figure 2a shows the results from the XWT significance testing between the NAOI and the
136 groundwater level records. Results are displayed as contours showing the percentages
of sites that exhibited a significant (0.05 a) XWP within the time-frequency spectrum. There
are five localised regions within the NAOI x GWL XWP spectrum that denote a wide-spread
significance between the GWL records and the NAOI. The greatest significance contours of
these regions (referred to here as focal points (FPs)) are labelled on figure 2a as: FP 1: 1934



at the 4.2 years periodicity (80% of records); FP 2: 1974 at the 8.5 years periodicity (40% of
records); FP 3: 1995 at 5.4 years (80% of records); FP 4: 2005 at 7 years (90% of records)
and; FP 5: 2012 at 2.9 years (60% of records).
These focal points are grouped into three larger regions within the 10% contour; between
1933 – 1940 spanning the 3- to 5-year periodicity; 1964 – 2020 spanning the 4- to 12-year
periodicity and; 2007 – 2017 spanning the 2- to 4-year periodicity. There is a single peak in
the time-averaged percentage plots (figure 2b) at the 7.5-year periodicity (average of 26% of
records)
Figure 2c shows the results from the CWT of the groundwater drought series (shown in Fig
2e). There are five regions of significant wavelet power in the groundwater drought
frequency spectrum that are labelled in figure 2c as follows; region 1: 1930 - 1950 in the 4-
to 8-year periodicity range (greatest power at 4.8 years); region 2: 1930 – 1945 in the 10- to
13-year periodicity range (greatest power at 11.7 years); region 3: 1960 – 1965 in the 2.5- to
3.5-year periodicity range (greatest power at 2.8 years); region 4: 1960 – 1990 centred at the
12- to 17-year periodicity range (greatest power at 15.4 years); and region 5: 1980 to 2020
at the 6- to 8-year periodicity range (greatest power at 7 years). There is a sixth significant
region starting in 2019 and covering periods between 2 and 5 years, however this is very
close to the end of the record and may be subject to edge effects. As such this region has
not been taken forward for discussion.
There are also two notable non-significant regions of medium strength wavelet power (>=
0.4); 1930 - 2000 at the 14- to 23-year periodicity range (centred at 16 years), and between
1960 and 1970 at the 8- to 16-year periodicity range (centred at 9 years). There are two
notable peaks in time-averaged wavelet power for the GWL drought series (figure 2d); the
greatest at the 7-year periodicity (average wavelet power of 0.38), and the second at the 14-
year periodicity (average wavelet power of 0.24).


Figure 3 shows the same as Figure 2 but for the streamflow (SF) case. There are six
localised regions within the NAOI x SF XWP spectrum that denote a wide-spread
significance between the SF records and the NAOI. FPs of these regions are labelled on
figure 2a; FP 1: 1940 at the 6.7-year periodicity (30% of records); FP 2: 1962 at the 5.2-year
periodicity (50% of records); FP 3: 1975 at the 8.5-year periodicity (40% of records); FP 4:
1994 at the 5.2-year periodicity (80% of records); FP 5: 2007 at the 7-year periodicity (90%
of records) and; FP 6: 2011 to 2015 at the 3.2-year periodicity (60% of records). These
centres are grouped into larger regions within the 10% contour; these are between 1933 –
1947 spanning the 5.5- to 8-year periodicity; 1960 – 1970 spanning the 4- to 8-year
periodicity; 1965 – 1990 spanning the 7- to 11-year periodicity; 1988 – 2000 spanning the 4-
to 5.5-year periodicity; 1995 – 2020 spanning the 4.5- to 11-year periodicity and 2007 –
2017 spanning the 2.5- to 4.5-year periodicity. There is a single peak in the time-averaged
percentage plots (figure 3b) at the 7.5-year periodicity (average of 29% of records)
Figure 3c shows the results from the CWT of the streamflow drought series (shown in Fig
3e). There are three regions of significant wavelet power in the groundwater drought
frequency spectrum that are labelled on Figure 3c; region 1: 1930 – 1935 in the 21 year
periodicity (this region appears clipped by the record start date, so the strongest wavelet
power for this region may not be captured); region 2: 1930 - 1937 in the 2.5- to 6.5-year
periodicity range (strongest power at 4.3 years) and; region 3: 1930 – 1960 in the 11- to 15-
year periodicity range (strongest power at 13 years);
There are four non-significant regions of medium strength wavelet power (>= 0.4); 1935 –
1945 at the 2- to 3-year periodicity; 1955 – 1965 at the 2- to 4-year periodicity; 1960 – 2015
at the 5.5- to 8-year periodicity; and 2000 – 2005 at the 2- to 5-year periodicity. The time-
averaged wavelet power for the SF drought series (figure 3d) contains multiple peaks
suggesting no dominant periodicity. The greatest peak is at the 7-year periodicity with an
average wavelet power of 0.21.
**4.2.    Cross-wavelet phase difference**




The cross-wavelet phase difference (ϕ) between water resource variables and the NAOI at the 7.5-year periodicity has been displayed in figure 4 for the GWL records and figure 5 for the streamflow records. The phase difference is a circular measurement where 0 indicates an in-phase relationship (analogous to zero lag) and +/- π indicates an out-of-phase relationship between the selected periodicity within the two variables (analogous to half a periodicity lag (3.75-years)). The purpose of these plots of phase differences are to visualise and understand the difference in phase between the NAO and water resources. Records have been split by their aquifer group in Figure 4, and by catchment region in figure 5, to understand if there are any general differences between regions.

The majority of groundwater level records cover the period 1970 to present, meaning general trends are more clearly presented for this time period. The phase difference of most GWL records can be defined by a sudden shift at approximately 1990 (figure 4). Values of ϕ generally range from between -1/4π and -3/4π (-0.76 to -2.36 rads; generally anti-phase) for the period 1975 to 1990 to between +1/4π and +3/4π (0.76 to 2.36 rads; generally in-phase) for the period 1990 to 2019 across all sites. This is with the exception of 17 sites across the South Chalk and Thames & Chiltern Chalk which have shorter ~anti-phase periods (between approximately 1985 and 1990). Average ϕ values for the period 1970 – 1990 (1990 – 2020) for each aquifer region are: -1.26 (1.41) in East Anglian Chalk; -2.25 (1.21) in Lincolnshire Chalk, 0.52 (0.83) in South Chalk, -1.37 (0.83) in Thames & Chiltern Chalk, 1.51 (1.21) in Greensands, -0.78 (0.66) in Limestone, -1.36 (1.09) in Oolite, -0.70 (1.35) in Sandstone. As such most aquifer regions experience an average reversal of polarity at 1990. Greensand GWL show no reversal when assessing average ϕ values, however 1 of the 3 sites in this aquifer group does show this reversal.

Similar to the GWL records, most SF records exhibit a shift in phase difference at approximately 1990, with catchment groups in the north of the UK showing minimal shifts (i.e., NW Scotland, E Scotland, NI, and NW England) (figure 5). In the southern catchment groups, values of ϕ generally range from between -1/2π and ±π (generally anti-phase) for

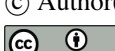



the period 1970-1990 (approximately prior to the shift) to between 0 and +3/4π (generally in-
phase) for the period 1990 to 2020 (approximately after the shift). Furthermore, catchment
groups in the east of the UK (i.e., E Scotland, NE England, East Anglia & SE England)
during the in-phase period (1990-2020) exhibit a notable transition to increased phase
difference (to approximately +3/4π) between 2000 and 2010 before decreasing to
approximately +1/4π in 2020.  Average ϕ values for the period 1970 – 1990 (1990 – 2020)
for each catchment region are: -0.21 (0.14) in North and West Scotland, 0.49 (0.86) in East
Scotland, -0.43 (0.46) in Northern Ireland, -0.44 (0.47) in NW England, 2.32 (1.08) in NE
England, 0.77 (0.64) in Wales and SE England, and 2.53 (0.99) in East Anglia and SE
England.
**4.3.    Modulation of dry season water resources**
Figure 6 shows two boxplots for each aquifer group, representing the distribution of mean (in
blue) and maximum (in red) dry-season GWL deviation as a result of the 7.5-year periodicity
(over the length of each of the record). Median values from each of these mean and
maximum boxplots are described below, and are referred to as med.mean and med.max
respectively.
The 7.5 year periodicity accounts for the greatest deviation of-dry season GWL in the Chalk
aquifer regions, with the Thames & Chiltern basin GWL showing the greatest modulation of
all groups showing med.mean of 0.94m  and a med.max of 1.38m. Two other Chalk groups
showed similarly strong modulations; the South Chalk basin GWL (med.mean: 0.7m,
med.max: 1.07m); and the Lincolnshire Chalk GWL (med.mean:.56m, med.max: 0.77m).
The East Anglia GWL show lowest modulation of the Chalk (med.mean: 0.16m, med.max:
0.34m), similar to GWL in the Limestone (med.mean: 0.35m, med.max: 0.51m) and the
Oolite (med.mean: 0.21m, med.max: 0.33m). Lowest overall modulations are found in the
Sandstone (med.mean: 0.15m, med.max: 0.25m) and Greensands aquifers (med.mean:
0.12m, med.max: 0.17m).





Figure 7 shows the same as figure 6 but for the streamflow case. Streamflow modulations
are measured as relative to the standard deviation of each record. Modulation of streamflow
for each catchment group are (in descending order of med.mean); Wales & south-west
England (med.mean: 0.32, med.max: 0.50);  East Anglia & south-east England (med.mean:
0.31, med.max: 0.53); Northern Ireland (med.mean: 0.29, med.max: 0.50); West Scotland
(med.mean: 0.27, med.max: 0.46); north-east England (med.mean: 0.27, med.max: 0.47),
north-west England (med.mean: 0.26, med.max: 0.46), east Scotland (med.mean: 0.21,
med.max: 0.39).





Figure 2 – a) Significance (95% CI) contours between GWL and NAOI, b) time-averaged
proportion of gwl records with a significant XWP with the NAOI (measured as a decimal
fraction), c) wavelet (spectral) power of GWL drought series, d) time-averaged wavelet
(spectral) power of GWL drought series, e) GWL drought coverage time series, f) temporal
coverage of records.



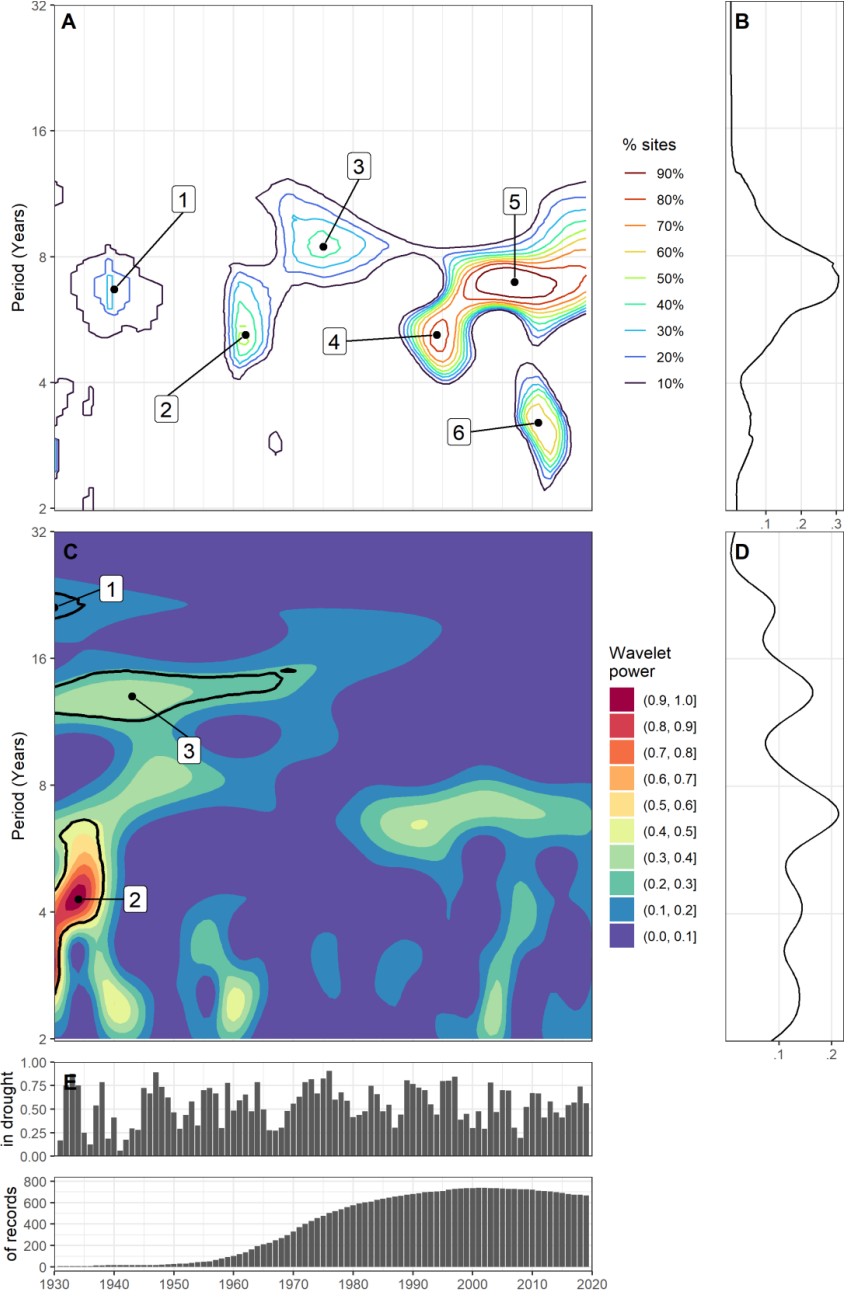

Figure 3 – a) Significance (95% CI) contours between SF and NAOI, b) time-averaged
proportion of SF records with a significant XWP with the NAOI (measured as a decimal
fraction),c) wavelet (spectral) power of SF drought series, d) time-averaged wavelet
(spectral) power of SF drought series, e) SF drought series showing proportion of records in
drought each year, f) temporal coverage of records.



441

Figure 4 – Phase difference between the NAOI and each GWL record for the GWL record
period. Results are grouped by aquifer regions. ϕ = 0 is equivalent to an in-phase
relationship and ϕ = ±π is equivalent to an antiphase relationship.

445





Figure 5 – Phase difference between the NAOI and each streamflow record for the
streamflow record period. Results are grouped by regions. ϕ = 0 is equivalent to an in-phase
relationship and ϕ = ±π is equivalent to an antiphase relationship.



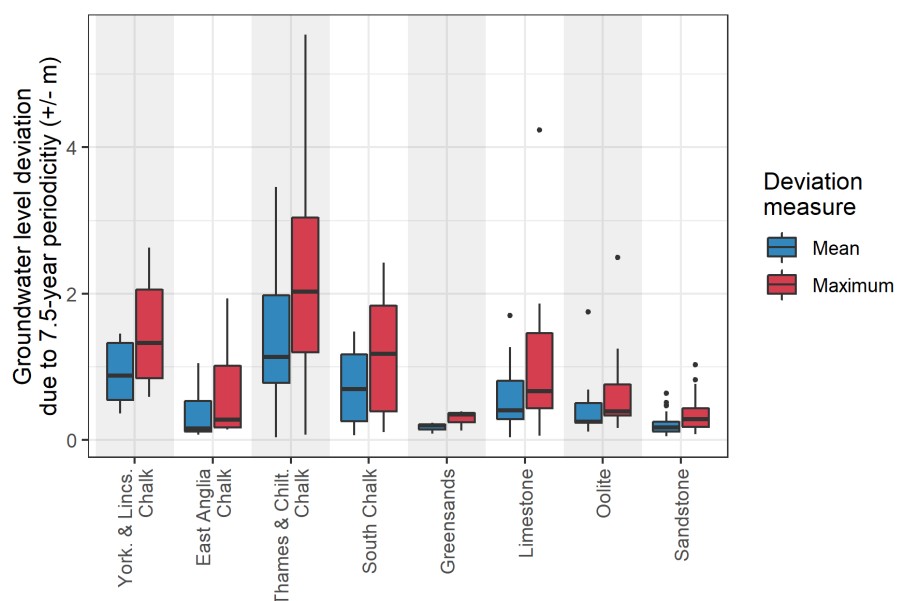


Figure 6 – Distribution of absolute mean and maximum modulation of summer groundwater

level as a result of the principal cross-wavelet periodicity between the NAOI and winter

Groundwater level by aquifer region

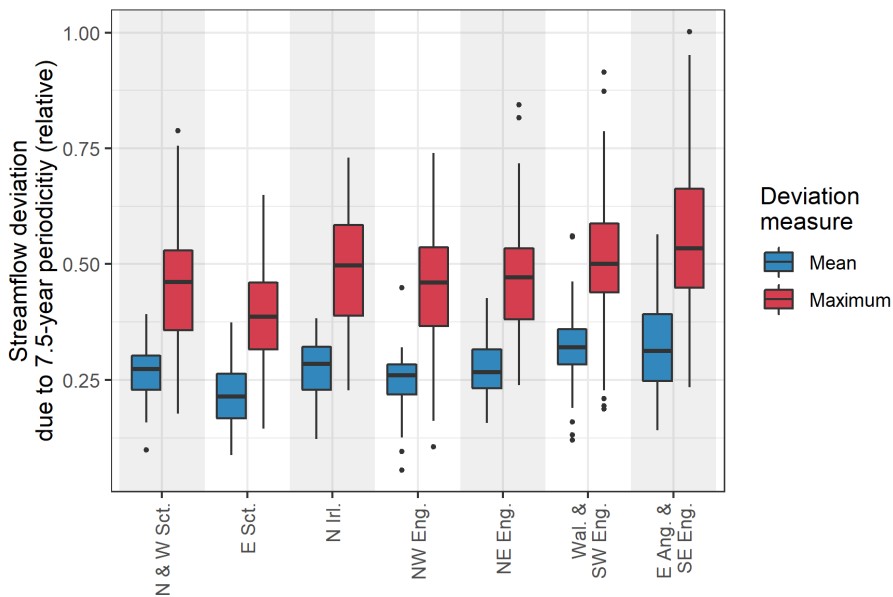






Figure 7 – Modulation of summer streamflow (relative to record standard deviation) as a
result of the principal cross-wavelet periodicity between the NAOI and winter streamflow.
**5. Discussion**
**5.1. Historical covariances between the NAOI and water resources at multiannual**
**periodicities**
Our results show that the dominant mode of multiannual covariance between the NAOI and
UK water resources is at the ~7.5-year periodicity. This is apparent in the time-averaged
covariance significance plots for groundwater (figure 2b) and streamflow (figure 3b). The
same 7.5-year periodicity is also the strongest average mode of periodic behaviour in water
resource extremes. Periodicities of similar lengths have previously been detected in
European GWL records, such as those in the UK (Rust et al, 2018 Holman et al, 2011),
Hungary (Garamhegyi et al, 2016), Spain (Luque-Espinar et al, 2008), Italy (De Vita et al
2011), and Germany (Liesch and Wunsch, 2019); and European streamflow records, for
example in the UK (Rust et al 2021; Burt and Howden, 2013) and Sweden (Uvo et al, 2021).
Our results therefore are consistent with principal periodicities detected in wider European
water resources and highlight the NAO's wide-scale control on water resource extremes.
Despite the prominence of the average 7.5-year periodicity in water resource variables, the
wider time-frequency spectra show that the NAO's multiannual control on water resources is
subject to considerable transience and non-stationarity across time and frequency. For
instance, the percentage of water resource records with a significant covariance with the
NAOI at the 7.5-year periodicity remains below 10% until between 1960 and 1965, with
significance becoming abruptly widespread (> 30%) between 1980 and 1985. As such this
suggests that the NAO's control on water resources, at the 7.5-year periodicity, has only
been prominent over the past four to five decades. Furthermore, prior to this mode of
behaviour, an approximate 16-year periodicity predominated the water resource extremes
record that did not covary with NAOI. Previous studies have associated a minimum in this
16-year cycle in water resources with the wide-scale 1976 drought (Rust et al, 2019) that


affected most UK water resources, particularly in the south of the country (Rodda and
Marsh, 2011). These findings are also consistent with Barker et al (2019) who demonstrate
longer duration drought events in the UK for the period 1940 to 1980 (approximately), and
comparatively shorter drought durations for the period 1980 to present. This may be
explained by a more prominent low-frequency influence on water resources and extremes
during this former period (1940 – 1980), causing longer negative anomalies on drought
indices. Finally, Holman et al (2011) linked a 16-year periodic behaviour in groundwater
records with the East Atlantic pattern, the second-most dominant mode of atmospheric
variability in the North Atlantic region. Our results could be interpreted as suggesting an
abrupt shift towards increased frequency of water resource extremes around 1970 to 1980
as a result of a transition of periodic control from the EA to the NAO. This interpretation may
expand on findings from Neves et al (2019) who demonstrate that historical droughts in
southwest Europe are better explained with a combination of NAO and EA influence.

Multiple studies have noted a marked change in European hydrological drought trends since
the 1970s, often in the context of the ongoing effects of climate change on water resources
(Tanguy et al 2021; Rodda and Marsh, 2011; Bloomfield et al., 2019). These impacts vary
depending on the water resource and region but can include changing drought frequency
(Spinoni et al, 2015; Bloomfield et al., 2019; Chiang et al, 2021), severity (Hanel et al, 2018;
Bloomfield et al., 2019), and increasing divergence of drought characteristic across Europe
(Cammalleri et al, 2020). We show here that a dominant 7.5-year periodicity, driven by the
NAO, has occurred coincident to these reported changing trends, and proceeded a
secondary periodicity of approximately 16 years. As such our results suggest that some of
the change in drought frequency that has been noted to have occurred since the 1970s, may
be in-part driven by the NAO's increased periodic control on water resources. Hydroclimate
studies often highlight that the interaction between climate change, ocean-atmosphere
processes and land-surface processes may be complex, resulting in non-linear hydrological



responses to increasing global temperatures (Rial et al 2004, Wu et al, 2018). As such, the
abrupt emergence of a 7.5-year periodicity between the NAO and water resource extremes
between 1980 and 1985, and its weaking since 2005, may be evidence of this type of non-
linear response. While there have been many studies assessing the impact of climate
change projections on the NAO (e.g. Rind et al (2005); Woolings and Blackburn (2012)),
there have been few that have investigated potential interactions between climate change
and multiannual periodicities in the NAO. As such, the role of climate change in affecting the
non-stationary periodicities (detected in this study) is currently unknown.
Yuan et al (2017) highlight the importance of suitable calibration period selection for the
development of drought early warning systems, particularly in climate change scenarios.
Many of these systems in Europe (e.g. Hall and Hanna, 2018; Svensson et al., 2015) rely on
high-resolution hydrometeorological datasets for calibration of historical relationships, many
of which are only available for recent decades (Rust et al, 2021b, Sun et al 2018). We show
here that frequency statistics potentially used as calibration bases for water resource early
warning systems can exhibit both multidecadal periods of stability and abrupt sub-decadal
non-stationarities, driven by multiannual behaviours in the NAO. Furthermore, we show a
weakening of the dominant 7.5-year periodicity since 2005, suggesting a different frequency
structure may predominate water resource extremes from the 2020s. This further highlights
the need for continuous recalibration of critical forecasting utilities, and the potential benefit
of including the NAOI as a covariate when understanding multiannual periodic variability in
European water resources.



### 5.2. Phase difference between NAO and water resource records at 7.5-year periodicity

The quantification of lead times between meteorological processes and water resource response is critical in the development of early warning systems for water resource management. As such, hydroclimate studies have sought to investigate temporal lags between multiannual periodicities in the NAO and water resource variables across Europe (Uvo et al, 2021, Neves et al 2019, Holman et al 2011). However, previous research has highlighted that the relationship strength and sign between the NAO and European rainfall is non-stationary at sub-decadal to decadal timescales (Rust et al 2021, Vicente-Serrano & López-Moreno, 2008). The extent to which this non-stationarity is projected to multiannual periodicities in water resources was previously unknown. Sign change is synonymous with a phase difference shift of approximately π between periodic components of the NAO and water resources, and as such has the potential to disrupt the projection of lead times into future scenarios. Here we assess the phase difference between the NAO and water resources at a country scale to identify the extent to which this non-stationary is present at multiannual periodicities.

Most water resources records exhibit an abrupt shift in phase difference of approximately -π around 1990. An earlier shift (of approximately +π) is also apparent between 1970 and 1980, however this is less temporally aligned across the fewer records that cover this period. This suggests that, for the period of approximately 1970 to 1990, the relationship sign between the NAO and water resources was inverted. Furthermore, the timing of this period of inversion generally aligns with reported periods of sign inversion in existing studies between the NAO and UK rainfall (Rust et al 2021, Vicente-Serrano & López-Moreno, 2008). It is interesting to note that this period of inversion is notably shorter for some groundwater level records of the Chalk (e.g., those in South Chalk and Thames and Chiltern Chalk). Rust et al (2021) showed the south and south east of the UK was subject to the increased non-stationarity of the NAO-precipitation relationship when compared to other regions, which



may explain these relatively short periods of relationship inversion. A similar spatial pattern
is shown in the streamflow records, with minimal phase difference shifts in northwest
England, Scotland, and Northern Ireland where more stable signs have been found by Rust
et al (2021b).
Localisation of this non-stationarity between the NAO and water resources at multiannual
periodicities suggests it is possible to identify a discrete time period of sufficient stationarity
from which to calculate lead-in times for early warning systems (for instance, between 1990
and 2020). However, phase differences for this period also show a degree of non-
stationarity, varying by up to approximately ±¼π. Some of this variance may be due to
changing storage dynamics within a catchment over time (Rust et al, 2014; Beverly and
Hocking, 2012), but also the introduction of red noise from reconstructing from non-
significant wavelets. This also explains the increased variance seen in aquifer groups
characterised by higher autocorrelation (e.g., Sandstone) (Bloomfield and Marchant, 2013),
and the relatively low variance seen in streamflow records which often have lower
autocorrelation when compared to groundwater level (Hannaford et al, 2021).   While this
can be minimised by calculating phase difference from significant wavelets only, we have
shown in the previous section that the significance between the NAO and water resources
and multiannual periodicities is also subject to notable non-stationarity.
Finally, in order to calculate accurate lead-in times between periodicities in the NAO and
water resources in future scenarios, a sufficient systematic understanding of the NAO sign
non-stationarity is required. However, there is limited research that has investigated the
causes for these modes of multiannual non-stationarity. Vicente-Serrano & López-
Moreno (2008) suggest that an eastward shift of the NAO's southern centre of action may
account for a portion of this variability, but highlight that further work is required for this to be
a sufficient explanation of a changing correlation between the NAO and European rainfall.
As such, existing non-stationarities between the NAO and water resources at multiannual





periodicities remains a considerable barrier to its application in improving preparedness for
future water resource extremes.

**5.3. NAO multiannual modulations on water resources in future scenarios**

Water resource management systems are in place across Europe to improve planning and
preparedness for the projected effects of climate change. As such, in order for multiannual
NAO modulations of water resources to have sufficient utility for water management systems
in future scenarios, they need to exhibit a comparable influence on water resources to the
projected effects of climate change. Here, we present historical modulations of summer
water resource variables from the principal NAO periodicity alongside expected impacts on
water resources from climate change projections in order to discuss their comparative
influence.
Jackson et al (2015) estimated median groundwater level change due to climate change in
24 boreholes across Chalk, limestone, sandstone and greensand aquifer groups in the UK
for the 2050s under a high emission scenario for September (as a typical annual minima of
groundwater levels in the UK). Median level from each site in Jackston et al (2015) have
been regrouped and averaged across the broad aquifer groups used in this study to allow
comparison with historical deviations in water resource results as a result of the NAO's 7.5-
year periodicity. This comparison is provided in Table 1. A mapping table of this comparison
is available in the supplementary material.



| Aquifer group | 50th %ile gwl change due to climate change ( m) | Gwl deviation due to 7.5-year NAO periodicity (± m) (med.mean) | Gwl deviation due to 7.5-year NAO periodicity (± m) (med.max) |
|---|---|---|---|
| Chalk (East Anglia) | -0.21 | 0.16 | 0.31 |
| Chalk (Lincolnshire) | -0.31 | 0.71 | 1.03 |
| Chalk (South) | -0.64 | 0.73 | 1.08 |
| Chalk (Thames / Chilterns) | -0.69 | 0.86 | 1.33 |
| Limestone | -0.28 | 0.35 | 0.51 |
| Oolite | -0.36 | 0.21 | 0.33 |
| Sandstone | -0.07 | 0.15 | 0.25 |
| Greensands | -0.10 | 0.12 | 0.17 |

Table 1 – synthesis of Table 3 from Jackson et al (2015). Median results from the absolute
teleconnection modulation on groundwater level from Figure 3 of this paper are also
presented for the mean and maximum modulation cases. NAO teleconnection modulations
greater than the reported 50th percentile climate change modulation are shaded in grey.
Historical modulations in groundwater level due to multiannual periodicities in the NAO were
greater than projected GWL modulation from a high emissions climate change scenario, in
all but two aquifer groups for mean NAO modulation (East Anglia Chalk, Oolite), and all but
one for maximum NAO modulation (Oolite). Similar degrees of GWL modulation from climate
change scenarios have been shown for wider European aquifer systems (e.g., Dams et al,
2011), and our results for NAO modulations of GWL are of a similar degree to those reported
by Neves et al (2019) for aquifers in the Iberian Peninsula. While few studies have looked at
multiannual NAO modulations of groundwater level across Europe, our results here suggest
a similar response across Western Europe, where the NAO has a greater influence on
precipitation (Trigo et al, 2002). However, existing studies notable uncertainties in the future
trends of groundwater level change due to climate change. For instance, Yusoff et al. (2002)
demonstrated that it was not possible to predict whether groundwater level would rise or fall
between 2020s and 2050s, Bloomfield et al. (2003) showed that groundwater levels were
expected to rise in the 2020s but fall in the 2050s, and, Jackson et al (2015) showed


reductions in annual and average summer levels but increases in average winter levels by
the 2050s. For streamflow, Kay et al (2020) give estimated modulations to low flows (Q95)
as a result of climate change (2050 horizon). While no Scottish catchments were used in the
study, percentage modulations for low flows were found to be mostly between 0 to -20%
change with some catchments showing up to -40% change for catchments in the West and
South West of the UK. Schnieder et al (2013) show similar low flow modulations across
Europe as a result of climate change, ranging from +20% for northwest Europe to -40% in
the Iberian Peninsula. As such, our results for streamflow (Figure 7) indicate that multiannual
NAO modulation of streamflow has been, on average, comparable to the expected change
due to climate change scenarios. NAO modulations in streamflow are notably less than
those found in groundwater level, as may be expected given the established sensitivity of
groundwater processes to long-term changes in meteorological fluxes (Forootan et al., 2018;
Van Loon, 2015; Folland et al., 2015).
Given the scale of multiannual NAO influence on water resource compared to the estimated
effects of climate change, the NAO may have the potential to impact the projected trend of
water resource variability in certain future scenarios more than was previously understood,
and therefore effect the required adaptive management response. However, existing
research has shown that that current GCMs do not fully replicate low frequency behaviours
in the NAO that have been historical recorded (Eade et al, 2021). Given the importance of
multiannual periodicities the NAO in defining water resource behaviour, demonstrated here
and in other research (e.g., Uvo et al, 2021; Neves et al, 2019), this raises notable
uncertainties in the use of GCMs outputs for projecting European water resource behaviour
into future scenarios. Findings reported here suggest that current projections from these
GCMs may contain error that is comparable to the current projected effect of climate change
on water resources. This therefore highlights the need for improved low frequency
representation in GCMs, and for an understanding of the non-stationary atmospheric
behaviours are can considerably influence wide-scale water resource behaviour.



Rust et al (2018) set out a conceptual model for how multiannual modulations of water
resources due to the NAO may provide a system for improving water resource forecasts and
management regimes. This model highlights the need for a systematic understanding of how
multiannual periodicities affect water resources over time, including temporal lags and
amplitude modulation between the NAO and water resources. We demonstrate that the
degree to which the NAO's 7.5-year periodicity has modulated historical water resources is
of a similar order of magnitude to the estimated impacts on water resource variables from
climate change projections. These results further show the importance of including the
influence of multiannual NAO periodicities on water resources in the understanding of future
extremes, as they have the potential to affect the required management regime for certain
resources in climate change scenarios. However, we also show that there are notable non-
stationarities in NAO periodicities over time and their relationship with water resource
response, for which there is limited systematic understanding in existing hydroclimate
literature.

## 669     6.  Conclusions

This paper assesses the utility of the relationship between the NAO and water resources, at
multiannual periodicities, for improving preparedness of water resource extremes in Europe.
We review this relationship in the context of non-stationary dynamics within the NAO and its
control on UK meteorological variables, as well as its potential impact on water resources in
climate change scenarios. We provide new evidence for the time-frequency relationship
between the NAO and water resources in western Europe showing that a wide-spread 7.5-
year periodicity, which predominates the multiannual frequency structure of many European
water resources, is the result of a non-stationary control from the NAO between
approximately 1970 and 2020. Furthermore, we show that known non-stationarities of the
relationship sign between the NAOI and European rainfall at the annual scale are present in
water resources at multiannual scales. A current lack of systematic understanding of both



these forms of non-stationarity, in existing atmospheric or meteorological literature, is a
considerable barrier to the application of this multiannual relationship for improving
preparedness for future water resource extremes. However, we also show that the degree of
modulation from multiannual NAO periodicities on water resources can be comparable to
modulations from a worst-case climate change scenario. As such multiannual periodicities
offer a valuable explanatory variable for ongoing water resource behaviour that have the
potential to heavily impact the required management regimes for individual resources in
climate change scenarios. Therefore, we highlight knowledge gaps in atmospheric research
(e.g. the ability of climate models to simulate NAO non-stationarities) that need to be
addressed in order for multiannual NAO periodicities to be used in improving early warning
systems or improving preparedness for water resource extremes.
**Data availability.**
The groundwater level data used in the study are from the WellMaster Database in the
National Groundwater Level Archive of the British Geological Survey. The data are available
under     license     from     the     British     Geological     Survey     at     https:
//www.bgs.ac.uk/products/hydrogeology/WellMaster.html (last accessed: 24/10/2021).
The streamflow data as well as the metadata used in this study are freely available at the
NRFA website at http://nrfa.ceh.ac.uk/ (last accessed: 25/10/2021).
The   data   that   support   the   findings   of   this   study   are   available   in   CORD   at
10.17862/cranfield.rd.16866868.  This study was a re-analysis of existing data that are
publicly available from NCAR at https://climatedataguide.ucar.edu/climate-data.

**Author contributions.**
WR designed the methodology and carried them out with supervision from all co-authors. WR
prepared the article with contributions from all co-authors.



**Competing interests.**

The authors declare that they have no conflict of interest.

**Acknowledgements.**

This work was supported by the Natural Environment Research Council (grant numbers NE/M009009/1 and NE/L010070/1) and the British Geological Survey (Natural Environment Research Council). JPB publishes with the permission of the Executive Director, British Geological Survey (NERC). MOC gratefully acknowledges funding for an Independent Research Fellowship from the UK Natural Environment Research Council (NE/P017819/1). We thank Angi Rosch and Harald Schmidbauer for making their wavelet package "WaveletComp" freely available.

**Financial support.**

This research has been supported by the Natural Environment Research Council (grant nos. NE/M009009/1 and NE/L010070/1), and MOC has been supported by an Independent Research Fellowship from the UK Natural Environment Research Council (NE/P017819/1).

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
