# Peer review of "The importance of non-stationary multiannual periodicities in the NAO index for forecasting"

_Hydrology and Earth System Sciences, 2021_

## Referee Comment (RC2)

[referee-annotated manuscript omitted]

---

## Author Response (AR1)

**Anonymous Referee #1 Comments**

We would like to thank Anonymous Referee #1 for their detailed review comments. We found them to be insightful, and, through our responses to them set out below, we believe that they have resulted in a much-improved paper. Below are the general and specific comments from Anonymous Referee #1, along with our responses to each comment.

**General Comment #1**: The record length looks like around 40 years on average, is it relatively short for periodicity analysis?

**Response to General Comment #1**: The continuous wavelet transform provides an instantaneous measure of periodicity strength within a time series and, as such, does not necessarily require data lengths that are multiple times longer than the periodicity being examined, unlike singular spectrum analysis, for example. We agree that this could be more clearly stated in the methodology section, and that caveats should be put into the discussion when considering the longer periodicities (for instance the ~ 16-year periodicity). However, the primary focus of this paper is the ~7.5 year periodicity, and given the typically 40 year record length and the instantaneous nature of the wavelet transform, we believe that the results and conclusions are valid.

Text has been amended at lines 189 – 196 and 525 – 528 to address this comment.

**General Comment #2**: Many of the stations are located in heavily urbanized regions, which means they may have significant artificial influence such as ground and surface water abstraction, effluent return, river regulation, and impounding reservoir (introduce another layer of uncertainty on the top of observation uncertainty). By including or excluding these stations might give very different results.

**Response to General Comment #2**: Over the period of analysis there have been both changes in regulatory and water resource management practices and that the latter will not have been applied in a consistent manner over all the catchments. Given this we might expect anthropogenic effects to add noise to the observations, but there is no reason to expect that they should impart a systematic signal or bias to the data and so systematically effect the observations or results. Counter to this, there is a precedent in literature for exacerbation of climatic cycles by anthropogenic processes which may affect the amplitude of the annual cycle but have little impact on multiannual cycles

We will add text to discussion section to highlight both of these points, and further highlight that future work should be undertaken on near-natural catchments to compare with the results here to better quantify the potential impacts of these processes.

We have added text at lines 685 – 696 to highlight that periodicities this study may include anthropogenic influences, and have identified this as an area of future research. This has been included to respond to the Referee comment here, and to the Editor's comment for which we are thankful

**General Comment #3**: It might be better to integrate the results section and discussion section since they are closely interlinked.

**Response to General Comment #3**: Although there may be some benefit in integrating the results and discussion sections, we believe there is a considerable degree of digestion required of the results 'as a whole' in order to draw out the key discussion points, which

might be lost or become unclear if these two sections were to be combined. We have therefore kept them separate.

**Specific Comment #1:** L2 (& L73-76): in the title, water resource extremes could be interpreted as flood AND drought, however, the paper only addressed drought. The title does not correctly reflect the contents of the paper.

**Response to Specific Comment #1:** We agree with this observation and propose changing "water resource extremes" in the title to "water resource drought". We also propose to remove the sentence addressing this between Lines 73 and 76.

We have adjusted the title, and have made edits at the following lines to remove reference to extremes and correctly focus on droughts; 73, 100, 112, 209, 210, 211, 227, 242, 268, 493, 500, 510, 517, 522, 545, 560, 619, 711, 719, 731, 739.

**Specific Comment #2:** L16: 'particularly in Europe'? – this also applies to other regions as indicated in many literatures.

**Response to Comment #3:** We agree that this opening sentence is unclear and propose changing to "Drought forecasting and early warning systems for water resource extremes are increasingly important tools in water resource management in Europe, where increased population density and climate change are expected to place greater pressures on water supply"

We have edited line 16 to make this narrative clearer.

**Specific Comment #4:** L159-161: might consider the UKBN dataset – a subset of NRFA stations that were considered near-natural with minimal human influence?

Harrigan, S., Hannaford, J., Muchan, K., & Marsh, T. J. (2018). Designation and trend analysis of the updated UK Benchmark Network of river flow stations: the UKBN2 dataset. Hydrology Research, 49(2), 552-567.

OR might use Factors Affecting Runoff (F.A.R.) codes published on the NRFA website to exclude stations that have huge human influence?

https://nrfa.ceh.ac.uk/content/catchment-summary-information

**Response to Specific Comment #4:** We did consider using the UKBN dataset but the limited number of gauges (146) leaves regions of the UK with relatively sparse coverage. Furthermore, we anticipate, that while there may be some exacerbation of periodicity strength as a result of climate-induced abstractions, other processes such as effluent discharge or river regulation are likely to introduce noise to the data and not alter the frequency distribution. We do however acknowledge that this is not clear in the existing text and will add new text into the discussion to address the potential influence of anthropogenic activities.

We have added text at lines 685 – 696 as per previous General Comment #2

**Specific Comment #5:** L168: the available period for NAOI is 1899-2021?

**Response to Specific Comment #5:** This is a typing error and will be corrected to 1899.

This error has been corrected at line 174.

**Specific Comment #6**: L201: do you mean Eq.1 here? Please check the equation numbers throughout the paper.

**Response to Specific Comment #6**: We agree that line 201 is unclear and would propose to change this to "…methodology given by equation 5 from Peters (2003)"

Text at line 215 has been corrected as per comment response, and equation numbers have been corrected at lines 243, 250, 269, 270, 290, 308.

**Specific Comment #7:** L232: what wavelet power can tell? Please clarify.

**Response to Specific Comment #7:** The power is used as an absolute measure of strength of the frequency spectrum, for ease of comparison across the continuous wavelet spectrum. We proposed to add text in around Line 232 to explain the purpose of an absolute measure.

Text has been added at lines 253 to clarify the reason for power use

**Specific Comment #8:** L242: punctuation mark is missing.

**Response to Specific Comment #8:** This typing error will be corrected.

Corrected at line 266.

**Specific Comment #9:** L367: it's not clear why 7.5-year periodicity is selected here, though the reason was provided in section 5, could consider refining the paper structure.

**Response to Specific Comment #9:** Agreed. We will add text in at Line 367 to specify that the 7.5-year periodicity is dominant across the period assessed.

We have added text at lines 392 to make the reasoning for 7.5-year period selection clear.

**Specific Comment #10:** L429 & 435: 'F' is not shown in Figures 2 and 3, and 'E' is not visible in Figure 3.

**Response to Specific Comment #10:** This figure will be corrected to ensure the 'E' and 'F' labels are visible

Correct 'F' Label added to figures, and 'E' on figure 4 has been made clearer.

**Specific Comment #11:** L594: could you please justify why choose the summer season?

**Response to Specific Comment #11:** Summer months have been selected in order to best capture the driest part of water resources annual cycle across a broad range of water resource records assessed in the study. We agree that this has been insufficiently explained and we will add text to the methodology section to ensure this is clear.

We have added text at lines 205-206, 256 – 258, 309 - 210 and 313-314 to make use of summer average clearer.

**Specific Comment #12**: L669: are there any limitations of the work worth acknowledging?

**Response to Specific Comment #12:** Based on previous comments, we agree that more acknowledgement is required of the potential impacts of anthropogenic influence on the water resource records used. As such, we will add text to the discussion to highlight this.

We have added text at lines 685 – 696 as per previous general comment #2

**Specific Comment #13:** L699: CORD means Cranfield Online Research Data? Please provide the expanded form.

**Response to Specific Comment #13:** Agreed, we will add the expanded form of this acronym.

We have added text as lines 747 – 748 to correct this.

**Referee #2 Comments.**

**Comment #1:** Greater clarity could be offered in the text in several places, the paper considers streamflow and groundwater series, not water resources, which includes additional data that are not discussed within the manuscript (e.g. lakes & reservoirs). In addition, it would be beneficial to be clear on the regional scope of the study, groundwater data from England (+Wales?) and streamflow data from the UK; in the abstract and conclusion inference is made to this being a European study, these need to be pulled back to UK, the clarity will help the reader.

**Response to Comment #1:** We agree that greater clarity is required when discussing water resources. We propose to add text to the introduction to make it clear that the water resources that are assessed and discussed throughout the paper are streamflow and groundwater, and not lakes and reservoirs.

Furthermore, we will add text to the introduction to make the regional scope of the study clear, i.e., that it is focusing on water resource in the UK, and we will adjust the text in the introduction and the conclusions to make it clear that this study fits into a wider European context but has only assessed data from the UK.

We have added text at lines 73 – 76 to clarify the scope of the term "Water Resources" throughout the paper, and we have amended text at lines 119, and 126 – 127 to make the geographical scope of this study clear.

**Comment #2:** The study focuses on low flows, not droughts, nor extremes as no discussion of high flow events. This requires clarification throughout.

**Response to Comment #2:** We agree that more wording is needed as to our definition of droughts used in the paper, which will inform the use of the term drought throughout the paper.

We have specified at line 73- 76 that the paper is focusing on water resource extremes rather than hydrological extremes (with definitions given), however we agree that the methods of the paper focus on droughts. As such we will update the title and text to consider water resource droughts rather than extremes. The title has also been modified in response to comments from Reviewer #1.

Regarding seasonal low flows (or low levels in the case of groundwater), while we are undertaking an assessment on low-flow / level values, we are also assessing multi-year anomalies within these. As such, we are considering multi-year below-average low flows are representative of hydrological drought. We agree that this aspect is not well addressed in the methods section, and we propose to add text here to improve this rationale.

We have amended text as lines 73, 100, 112, 209, 210, 211, 227, 242, 268, 493, 500, 510, 517, 522, 545, 560, 619, 711, 719, 731, 739 to focus the text on droughts rather than extremes. Additionally, text has been added to lines 234 – 237 to address our use of low flows in an assessment of multi-year drought.

**Comment #3:** Paragraph (lines 77-97) needs to make a clearer case for the relationship between NAO and streams/groundwater and multiannual periodicities, as the relationship between NAO and summer streamflow, when low flows are expected, is weak.

**Response to Comment #3:** We agree that more supporting literature could be included in this paragraph to better support the claims of a relationship between the NAO and low flows / droughts, at multiannual periodicities and how this differs from the annual-scale relationship. Additional text will be added to this paragraph to address this, along with improved supporting literature citations.

We have amended text at lines 90 – 93 and 965-967 to address this comment.

**Comment #4:** Aim and Objectives need revising, to reflect UK study, low flows not extremes and not studying European rainfall

**Response to Comment #4:** We agree that the aims and objectives could be clearer and more appropriate to the paper. As such we propose removing reference to "extremes" to focus on "droughts" (which will be previously defined), and reference to European rainfall will be removed.

As per previous responses, amendments have been made at lines 73, 100, 112, 209, 210, 211, 227, 242, 268, 493, 500, 510, 517, 522, 545, 560, 619, 711, 719, 731, 739 to focus the text on droughts rather than extremes. Additionally, text has been added to lines 232 – 235 to address our use of low flows in an assessment of multi-year drought.

**Comment #5:** In each case (streamflow and groundwater) 20-year series are included into the analysis, this study would be more robust if only longer series were included. 20 years is too short for multiannual analysis, consider >40 years.

**Response to Comment #5:** The continuous wavelet transform provides an instantaneous measure of periodicity strength within a time series and, as such, does not necessarily require data lengths that are multiple times longer than the periodicity being examined. We agree that this should be more clearly stated in the methodology section. However, the primary focus of this paper is the ~7.5 year periodicity, and given the 20-year minimum record length and the instantaneous nature of the wavelet transform, we believe that the results and conclusions are still valid.

Text amended at lines 189 – 196 and 525 - 528

**Comment #6:** You state there is no UK benchmark river flow series, there is, why not use this to overcome concerns you then note (https://doi.org/10.2166/nh.2017.058).

**Response to Specific Comment #6:** We did consider using the UKBN dataset, however since this represents 146 gauges this does leave some regions of the UK with relatively sparse coverage. Furthermore, we anticipate, that while there may be some exacerbation of periodicity strength as a result of climate-induced abstractions, other processes such as effluent discharge or river regulation, these are likely to introduce noise to the data and not alter the frequency distribution. We do however acknowledge that this is not clear in the

existing text and will add text into the discussion to address the potential influence of anthropogenic activities.

As per our responses to Referee #1, Comment #2, We have added text at lines 685 – 696 to highlight that periodicities this study may include anthropogenic influences, and have identified this as an area of future research. Additional citations and reference added at 962 – 964.

**Comment #7**: typo on NAOI length line 168

**Response to Comment #7:** This will be corrected

This has been corrected at line 174

**Comment #8:** The threshold sampling approach you apply does not identify droughts, but low flows. This might seem pedantic but is important. You need to present some indication of how many years are identified as low flow years for each station using this approach as low flow years.

**Response to Comment #8:** We agree that the distinction between low flows and droughts, in the case of streamflow, needs to be made clearer and text will be added to the methods section to address this. However, given general uncertainty in drought definition, we consider that the widely cited drought threshold methodology proposed in Peters (2003) is appropriate. In general, we are using this threshold approach to identify years in which streamflow are below a given threshold, and by extracting multiannual periodicities in this series, we identify multi-year periods of below-threshold streamflow.

We have included additional text at lines 234 – 237 to address this comment.

**Comment #9:** I am concerned by the grouping of the results in section 4.1 together. We know that droughts are regionally coherent and often impact regions rather than the whole country, this will impact on low flows. There is also a skew towards stations of longer length and greater density in the SE with reducing length and frequency as you move north, this will skew your findings. This section would be much better if it was undertaken regionally, as you demonstrate the phased relationship to NAO for streamflow (Fig 5) is regionally highly variable. I actually think this is a really interesting section and further exploration of the regional low flow-NAO periodicity would be of interest and insightful, this could then be discussed further and would allow a more nuanced understanding to be garnered of regional patterns. You might expand the discussion of NAO track shifts too, based on the regionalisation, as the stronger consistent signal in north would suggest that that the change is consistent with that postulated by Comas-Bru & McDermott (2013). You can then assess whether the 7.5 years is consistent across regions. This is a key section and frames much of the rest of the results and discussion, as from this section you select the 7.5 year periodicity. Greater consistency is identified in groundwater-NAO phases, but this really only covers England, with a strong skew to SE again.

**Response to Comment #9:** We agree that splitting the streamflow spectra regionally would provide some very interesting results, however the intention of the figures 2 and 3 are to show general, wide-spread behaviours between the NAOI and the two water resource variables. Whereas the strength of the 7.5-year periodicities is explored in figure 7 for streamflow, and the regional assessment is undertaken from this figure in the discussion. The maximum and mean deviations of streamflow as a result of the 7.5 year periodicities are also reported in order to minimise any effect of varying record lengths between regions used in this study. Figure 1 does show that difference in regional record length is minimal, with the

exception perhaps of Northern Ireland. Text has been added to the results section to make the purpose of these figures clear, as a measure of broad-scale water resource behaviour. Regarding the skew within the groundwater records (covering England and mostly the Chalk), we agree that there could be clearer text regarding the applicability of these results. As such cautionary text will be added to the discussion section to address this.

Regarding the influence of NAO track shifts, this is a well-made point and something that we have considered when preparing the discussion. However, as previous research has shown (Rust et al, 2021a), catchment processes affect a significant modulation on the strength of the 7-year periodicity in regional rainfall. As such, the signal presence in water resource records alone is insufficient to comment on regional NAO influences on meteorological variables.

Text to indicate the purpose of figures 2 and 3 has been added to line 329-330. Furthermore, we have added additional text at lines 697 – 701 to address the applicability of groundwater results and the skew towards England and the Chalk.

**References**

Peters, E. Propagation of drought through groundwater systems - illustrated in the Pang (UK) and Upper-Guadiana (ES) catchments. Ph. D. thesis, Wageningen University. 2003

Rust, W., Cuthbert, M., Bloomfield, J., Corstanje, R., Howden, N., and Holman, I.: Exploring the role of hydrological pathways in modulating multi-annual climate teleconnection periodicities from UK rainfall to streamflow, Hydrol. Earth Syst. Sci., 25, 2223–2237, https://doi.org/10.1016/j.jhydrol.2014.05.052, 2021a.